# OpenReview forum: "MACD: Model-Aware Contrastive Decoding via Counterfactual Data for Video-LLMs"
_ICML.cc/2026/Conference — ICML 2026 regular_

### Official Review · Reviewer_1Ef2 · 2026-02-25

**Soundness:** 3
**Presentation:** 4
**Significance:** 3
**Originality:** 3
**Overall Recommendation:** 5
**Confidence:** 4

**Summary:**

This paper proposes **MACD**, a training-free decoding strategy that improves contrastive decoding (CD) by replacing heuristic perturbations with **model-aware counterfactual inputs**. The method constructs object- and frame-level masks using an external detector, then optimizes mask strengths via gradient ascent to maximize query reconstruction loss. The resulting counterfactual view highlights evidence-critical regions specific to the model and query. By contrasting the original and optimized masked inputs during decoding, MACD suppresses tokens unsupported by visual evidence while preserving grounded ones. Experiments across multiple Video-LLM backbones and benchmarks demonstrate consistent reductions in hallucination and competitive or improved task accuracy.

**Compliance With Llm Reviewing Policy:**

Affirmed.

**Final Justification:**

The author addresses these concerns by emphasizing both robustness and evaluation breadth. Regarding query variability, the author notes that mask optimization remains generally consistent.
They also provide additional ablation studies that directly compare fix masks with VCD to better isolate performance gains and trade-offs.

**Key Questions For Authors:**

### Questions

1. Would alternative optimization objectives for mask construction yield stronger or more stable results?
2. How stable is mask optimization across different query types (descriptive vs reasoning-heavy, short vs long, simple vs compositional)?

### Suggestions

1. Provide deeper analysis on how different query types influence mask optimization and decoding behavior.
2. Include additional ablation studies directly comparing MACD variants with VCD.

**Limitations:**

yes

**Strengths And Weaknesses:**

### Strengths

1. The paper is well motivated by the limitations of random perturbation in prior CD methods and empirically demonstrates that model-aware masks provide more effective counterfactual views than heuristic degradations.
2. The gradient-based identification of critical frames and regions, followed by constructing optimized counterfactual inputs, is conceptually clear, intuitive, and coherently integrated into the contrastive decoding framework.
3. MACD is training-free, backbone-agnostic, and compatible with multiple decoding strategies (e.g., greedy, nucleus, self-consistency), making it practical.
4. The authors validate mask quality through human evaluation, showing that optimized masks better align with query-relevant evidence, supporting the claim that performance gains stem from high-quality, model-aware counterfactuals.

### Weaknesses

1. The method links query reconstruction loss to visual evidence importance (frames or regions), but lacks theoretical analysis or systematic empirical study on this connection. The effect of different query characteristics (length, difficulty, event-centric vs attribute-centric, reasoning depth) is largely unexplored.
2. MACD relies on external object detection (e.g., YOLO). Detection errors or biases may propagate into decoding.
3. The reported memory increase (Table 7) is non-trivial, which may limit scalability to long-video QA or resource-constrained deployment scenarios.
4. In Table 2, the variant without mask training underperforms VCD (Table 1) on recall, F1, and accuracy. The ablation analysis does not sufficiently explain why fixed masks degrade performance relative to random perturbations.

---

> ### Author Rebuttal · Authors · 2026-03-31
>
> **W1 & Q1/2 & Suggestion 1:**
> We clarify our claim and provide a systematic breakdown below.
> * **Reconstruction Loss vs. Evidence Importance (W1):** Maximizing target query loss is an operational measure of evidence sensitivity. If masking a region substantially increases query token loss, it likely contains evidence the frozen model relies on.
> * **Alternative Objectives (Q1):** Query reconstruction loss isn't the sole optimal objective, but we chose it because it is query-conditioned, label-free, and available at inference, making it practical for deployment.
> * **Systematic Study Across Query Types (W1, Q2, S1):** To study stability across query types, we break down EventHallusion (Qwen2.5-VL-3B) into descriptive vs. reasoning-heavy, attribute-centric vs. event-centric, and short vs. long queries. As shown below, MACD improves over both Baseline and VCD across all subgroups, with especially large gains on reasoning-heavy and event-centric subsets.
>
> | Query Dimension | Sub-category | Baseline Acc. | VCD Acc. | MACD Acc. |
> |---|---|---|---|---|
> | Reasoning Depth / Difficulty | Descriptive | 0.68 | 0.675 | 0.74 |
> | | Reasoning-heavy | 0.56 | 0.565 | 0.68 |
> | Content Focus | Attribute-centric | 0.65 | 0.645 | 0.71 |
> | | Event-centric | 0.59 | 0.595 | 0.71 |
> | Query Length | Short (<=Median Length) | 0.62 | 0.64 | 0.73 |
> | | Long (> Median Length) | 0.58 | 0.60 | 0.70 |
>
> **Deeper Analysis:** These subgroup results do not by themselves prove the mechanism, but they are consistent with the following pattern: model-aware masking is particularly helpful when the model is more vulnerable to language priors or spurious temporal correlations, while still remaining beneficial in simpler descriptive settings.
>
> **W2:** MACD does not require a perfect detector, only used to initialize candidate object regions. The final counterfactual view is constructed jointly from object-level and frame-level masks, with the mask strengths further refined by Video-LLM gradients.
> * **Failure Analysis & Coverage:** We sampled the test queries across all benchmarks. Only a marginal fraction (~1-2%) of queries involve entities that the detector fails to capture (typically amorphous elements like smoke or liquids).
> * **Robustness Beyond a Single Detector Setting:** We already include a detector-threshold robustness study in the appendix (Table 13). Across noisy, default, and strict settings, MACD remains highly effective, suggesting its robustness.
> * **Modularity Across Proposal Mechanism:** The core MACD objective relies on spatial proposals, not the specific detector architecture. Alternative detectors (e.g., Grounding DINO) can be seamlessly plugged in to handle these rare edge cases.
> * **Built-In Fallback:** When detectors miss entities, frame-level masks act as a detector-agnostic temporal fallback.
>
> **W3:**
> Under the main EventHallusion configuration, 1-step MACD increases latency from 409.11 ms to 554.47 ms (+35.5%) and peak GPU memory from 7.17 GB to 11.59 GB (+61.7%). Our point is that MACD cost is limited to a single-step, frozen-backbone optimization. Under a compute-constrained setting (280×280, 1 fps), the same 1-step variant still improves accuracy from 34.20% to 48.79% with 24.47% latency overhead.
> Regarding long-video QA, Video-MME specifically evaluates models on comprehensive, long-form temporal content. As shown in Table 1, the exact same 1-step MACD configuration operates seamlessly on Video-MME without memory bottlenecks, improved long-video understanding (e.g., Accuracy increases from 0.46 to 0.64 on Qwen3-VL-2B).
>
> **W4 & Suggestion 2:**
> To provide a clearer picture of our method's advantages, we have updated our ablation study to include VCD as a direct reference point.
>
> | Variant | Precision | Recall | F1 | Accuracy |
> |---|---|---|---|---|
> | VCD (Random Perturbation) | 0.69 | 0.68 | 0.68 | 0.62 |
> | No Mask Training (Fixed Mask) | 0.79 | 0.58 | 0.67 | 0.61 |
> | **MACD (Targeted Optimization)** | 0.8 | 0.82 | 0.81 | 0.83 |
>
> The "No Mask Training" performance drop is due to the difference between blind occlusion and random perturbation. A fixed mask blindly obscures detected objects; if an object is vital to the query, completely masking it destroys necessary visual context, forcing the model to guess, which severely degrades Recall (0.58) and Accuracy (0.61). In contrast, VCD applies global unstructured noise, which degrades overall image quality but often leaves scattered visual cues intact, allowing the model to retain partial context (achieving better Recall 0.68 and Accuracy 0.62). This comparison highlights that applying unoptimized object-level priors is detrimental (F1 drops to 0.67) because it blindly destroys context. MACD resolves this flaw by using a model-conditioned measure of evidence sensitivity. By leveraging target query loss gradients, our optimization selectively penalizes only the specific features driving hallucination while preserving valid contextual evidence (reaching F1 0.81 and Accuracy 0.80).

---

> > ### Author Rebuttal · Reviewer_1Ef2 · 2026-04-02
> >
> > Thank you for the detailed response and the additional experiments.

---

> > > ### Author Response · Authors · 2026-04-03
> > >
> > > Thank you for your feedback and for taking the time to review our revisions. We truly appreciate your recognition of the additional analyses and clarifications. Your suggestions helped us significantly improve the clarity and strength of the paper. We also sincerely appreciate your updated evaluation.

---

### Official Review · Reviewer_FELZ · 2026-03-09

**Soundness:** 3
**Presentation:** 3
**Significance:** 4
**Originality:** 3
**Overall Recommendation:** 4
**Confidence:** 3

**Summary:**

This manuscript proposes MACD, a model-aware contrastive decoding method that combines counterfactutal construction. The counterfactual views are integrated into constrastive decoding to suppress unsupported tokens during generation. Experiments are conducted on multiple VideoLLMs and benchmarks, demonstrating consistent improvement on hallucination reduction. The manuscript also discuss its limitation and social impacts.

**Compliance With Llm Reviewing Policy:**

Affirmed.

**Final Justification:**

The authors solved my concerns, so that I keep a positive score.

**Key Questions For Authors:**

1. How to ensure the detection accuracy of the detector? Will alternative detectors affect the performance of the proposed strategy?

2. Can the proposed method generalize to other video-related tasks, such as reasoning or long video understanding?

**Limitations:**

Yes

**Strengths And Weaknesses:**

Strengths:

1. The idea of generating counterfactual video inputs guided by the model's gradients is intuitive. This is a meaningful improvement over prior approaches in contrastive decoding.

2. The proposed strategy keeps the model frozen and only adds a forward pass, which is easy to integrate with existing VideoLLMs without retraining.

3. The experiment is comprehensive. The proposed strategy is evaluated on multiple mainstream models and benchmarks, and performs consistently well.

Weaknesses:

1. The proposed strategy relies on an external detector (e.g., YOLO) to obtain object masks. Detection failures or biases may propagate into the decoding process and affect robustness.

2. Experiments focus mainly on hallucination-related benchmarks. It remains unclear whether the method consistently preserves general reasoning or long-range temporal understanding in broader tasks.

3. Minor: The capitalization of section titles is inconsistent and should be unified throughout the manuscript. "Video language models" in line 12 should be "Video large language models".

---

> ### Author Rebuttal · Authors · 2026-03-31
>
> **W1 & Q1:** MACD does not require a perfect detector, nor is it tied to YOLOv11 as an oracle. In our framework, the detector is only used to initialize candidate object regions; the final counterfactual view is constructed jointly from object-level and frame-level masks, with the mask strengths further refined by Video-LLM gradients.
>
> * **Failure Analysis & Coverage:** We sampled the test queries across all of our benchmarks. We found that only a marginal fraction (~1-2%) of queries involve entities that the detector fails to capture (typically amorphous elements like smoke or liquids). Crucially, precise bounding-box localization for these rare edge cases is rarely necessary to answer the actual queries, as the broader visual context provides sufficient semantic clues.
> * **The Built-In Fallback:** MACD does not rely solely on detector-localized object regions. When the detector misses an entity or provides incomplete spatial proposals, the frame-level mask still offers a detector-agnostic temporal perturbation path. This allows the counterfactual branch to retain partial evidence mining ability rather than failing completely. Our ablations further support that the frame-level branch plays a complementary role in the full system.
> * **Robustness Beyond a Single Detector Setting:** We already include a detector-threshold robustness study in the appendix (Table 13). Across noisy, default, and strict settings, MACD remains highly effective, suggesting that the method is not brittle to a specific YOLOv11 confidence configuration.
> * **Modularity Across Proposal Mechanism:** The core MACD objective relies on spatial *proposals*, not the specific detector architecture. Alternative open-vocabulary detectors (e.g., Grounding DINO) can therefore be seamlessly plugged in to handle these rare edge cases, without changing the counterfactual optimization or contrastive decoding pipeline.
>
> We will explicitly add this empirical breakdown of unsupported query entities, the literature context regarding amorphous objects, and their corresponding performance subset in the revised manuscript.
>
>
> **W2 & Q2:**
> We appreciate the reviewer for raising this important question. We would like to clarify that MACD is not strictly a narrow "hallucination fix," but a general inference-time grounding mechanism. To demonstrate this, our main experiments already include comprehensive evaluations on general reasoning and long-video understanding benchmarks, where MACD shows consistent and significant improvements.
>
> Specifically, as detailed in Section 5.1 and Table 1:
>
> * **General Temporal Reasoning (MVBench & Perception Test):** MVBench evaluates a broad range of 20 temporal tasks requiring low-to-high-level cognition, while Perception Test measures diagnostic reasoning (memory, physics, semantics). Across various backbones, MACD consistently improves accuracy. For example, on Qwen3-VL-2B, MACD elevates MVBench accuracy from 0.55 (Baseline) to 0.77, and Perception Test accuracy from 0.55 to 0.62.
> * **Long-Video Understanding (Video-MME):** Video-MME explicitly evaluates comprehensive understanding across varying durations, including medium and long videos. On this benchmark, MACD yields highly robust gains, lifting Qwen3-VL-2B accuracy from 0.46 to 0.64, and Qwen2-VL-7B accuracy from 0.51 to 0.59.
>
> **Why MACD Generalizes:** Complex reasoning and long-video tasks heavily require the model to anchor its answers to specific visual evidence over time, rather than relying on language priors or spurious temporal co-occurrences. By optimizing spatial-temporal masks to locate task-critical evidence and using contrastive decoding to enforce grounded token selection, MACD naturally enhances the model's performance on these broader, context-heavy tasks.
>
> We will ensure that this generalization capability is highlighted more prominently in the introduction and discussion sections of the revised version.
>
> **W3:**
> We appreciate the reviewer for catching these details. In the revised manuscript, we will unify the capitalization of all section titles and correct "Video language models" to "Video large language models" in line 12. Furthermore, we will conduct a comprehensive proofreading pass to refine the grammar and polish the overall writing of the paper.

---

> > ### Author Rebuttal · Reviewer_FELZ · 2026-04-01
> >
> > Thanks for the rebuttal. I think my concerns have been well-addressed. I also thank the authors for honestly acknowledging some limitations in their method, though it does not affect the main contribution of the paper. I hope the authors would add these clarifications in the revision.

---

> > > ### Author Response · Authors · 2026-04-01
> > >
> > > Thank you for reviewing our rebuttal. We are glad to hear that your concerns have been fully addressed. As you suggested, we will make sure to include the clarifications and the discussion on limitations in the revised paper.
> > >
> > > Since you noted that the issues are resolved, we kindly ask if you would consider updating your score. Your support is very important to our submission. If there is anything else we can do to improve the paper, either during the rebuttal period or in the final revision, please let us know. We are happy to address any remaining points.

---

### Official Review · Reviewer_wVwH · 2026-03-10

**Soundness:** 3
**Presentation:** 2
**Significance:** 2
**Originality:** 2
**Overall Recommendation:** 4
**Confidence:** 4

**Summary:**

This paper proposes MACD (Model-aware Counterfactual Data based Contrastive Decoding), an inference-time method designed to reduce hallucinations in Video Large Language Models (Video-LLMs). Unlike conventional contrastive decoding approaches that rely on random perturbations such as Gaussian noise or frame dropping, MACD uses the model’s own loss gradients to identify query-relevant objects and frames that provide critical visual evidence. Based on this feedback, the method constructs object-level and frame-level counterfactual inputs by selectively masking those regions. The original video and the counterfactual video are then used together in a contrastive decoding framework, which suppresses tokens unsupported by visual evidence while preserving tokens grounded in the video content. Experiments on multiple benchmarks—including EventHallusion, MVBench, Perception-Test, and Video-MME—and across several backbone models from the Qwen and InternVL families show that MACD consistently reduces hallucinations while maintaining or improving task accuracy. The method is particularly effective in challenging scenarios involving small, occluded, or co-occurring objects. Importantly, MACD is training-free, keeps the backbone model frozen, and introduces only a small computational overhead by adding roughly one extra forward pass during decoding, making it practical for real-world deployment.

**Compliance With Llm Reviewing Policy:**

Affirmed.

**Final Justification:**

The authors have addressed most of my concerns.

**Key Questions For Authors:**

In my view, the proposed MACD does not feel significantly novel compared to existing approaches in data augmentation or contrastive decoding methods. The authors claim superiority over previous augmentation techniques that inject noise, but attempts to generate and utilize counterfactual data have already been explored in many areas beyond Video-LLMs. The paper is well organized and the authors’ targeting is understandable, but the novelty of the method needs to be explained and emphasized more clearly. If necessary, it would also be helpful to reference ideas from studies in other domains or models, even if they have not been previously applied to Video-LLMs.

**Limitations:**

Yes

**Strengths And Weaknesses:**

**Strengths**

1.The authors provide clear targeting through model feedback–based counterfactual data generation. Unlike existing methods that use random perturbations such as random noise or frame dropping, MACD utilizes gradient-based feedback from the model to identify and mask important objects and frames. Therefore, counterfactual data can be generated around regions that actually influence the model’s predictions, enabling more targeted suppression of hallucinations.

2.The precise approach that considers both object-level and frame-level information is effective. MACD does not simply perturb entire frames but uses both object-level and frame-level masking together. Through this, hallucinations can be reduced more precisely even in complex scenes where small objects, occluded objects, or multiple objects appear simultaneously.

3.Since the proposal operates at the inference stage without requiring training, it appears applicable to various models and tasks.
It has the practical advantage that it can be relatively easily applied to various already-trained Video-LLM models.

4.The method shows stable performance improvements across models of various sizes and benchmarks.
Demonstrating generalization beyond a specific model or dataset suggests strong scalability.


**Weaknesses**

1.It is highly dependent on the model’s object detection and tracking performance.Since the first step of the method is object detection and tracking, if the detection or tracking is inaccurate, incorrect masks may be generated, which can reduce the quality of the counterfactual data.

2.There is a large increase in computational cost during the inference stage. Although it is said that training is not required, gradient computation for mask optimization and an additional forward pass are needed, so the inference cost increases compared to basic decoding.

3.The biggest concern is that the proposed method basically uses an approach similar to existing contrastive augmentation techniques and contrastive decoding methods. While having clear targeting and showing definite improvements is an advantage, the approaches for solving the problem are similar to existing directions. Of course, improving and combining existing techniques can also be good research, but to me MACD did not feel particularly new.

---

> ### Author Rebuttal · Authors · 2026-03-31
>
> **W1:** MACD does not require a perfect detector, only used to initialize candidate object regions. The final counterfactual view is constructed jointly from object-level and frame-level masks, with the mask strengths further refined by Video-LLM gradients.
> * **Failure Analysis & Coverage:** We sampled the test queries across all of our benchmarks. We found that only a marginal fraction (~1-2%) of queries involve entities that the detector fails to capture (typically amorphous elements like smoke or liquids). Crucially, precise bounding-box localization for these rare edge cases is rarely necessary to answer the actual queries, as the broader visual context provides sufficient semantic clues.
> * **The Built-In Fallback:** MACD does not rely solely on detector-localized object regions. When the detector misses an entity or provides incomplete spatial proposals, the frame-level mask still offers a detector-agnostic temporal perturbation path. This allows the counterfactual branch to retain partial evidence mining ability rather than failing completely. Our ablations further support that the frame-level branch plays a complementary role in the full system.
> * **Robustness Beyond a Single Detector Setting:** We already include a detector-threshold robustness study in the appendix (Table 13). Across noisy, default, and strict settings, MACD remains highly effective, suggesting its robustness.
> * **Modularity Across Proposal Mechanism:** The core MACD objective relies on spatial proposals, not the specific detector architecture. Alternative detectors (e.g., Grounding DINO) can be seamlessly plugged in to handle these rare edge cases.
>
> **W2:** for inference efficiency. We use "lightweight" to distinguish MACD from methods requiring costly parameter fine-tuning or multiple full auto-regressive passes. As detailed in Appendix B.1 and B.2, the overhead is strictly constrained and highly practical, due to the following designs:
> * **Frozen LLM Parameters:** The backpropagation only updates the lightweight input mask tensor ($r$), while the Video-LLM remains frozen.
> * **1-Step Optimization Policy:** Crucially, MACD does not perform a lengthy optimization loop, but only a 1-step gradient ascent.
> * **Quantified Low Overhead:** Our rigorous profiling on an A100 GPU (Tables 6 & 7) demonstrates that a 1-step MACD adds only ~24% to 35% latency overhead.
>
> **W3:**
> We thank the reviewer for recognizing our clear targeting and strong empirical results. While contrastive decoding and counterfactuals are indeed broadly established concepts, MACD is not a mere combination of existing heuristic augmentations and CD. The core novelty of MACD is a fundamental paradigm shift: moving from *random, heuristic perturbations* to *model-aware, targeted counterfactual generation* guided by the Video-LLM's own feedback. We clarify this crucial distinction below:
>
> * **Random Perturbations vs. Model-Aware Discovery:** Existing visual CD methods rely on "random perturbations" (like global noise or patch masking) that cannot guarantee alignment with the visual cues actually driving hallucination. In contrast, MACD does not rely on guessing. By maximizing the target query loss via gradient ascent, MACD leverages the frozen Video-LLM's own feedback to mathematically pinpoint the exact regions responsible for the hallucination.
> * **Bridging Adversarial Feedback with Generative Decoding:** We appreciate the suggestion to reference broader domains. While counterfactuals exist in areas like NLP adversarial attacks or CV interpretability, MACD's novelty is structural. It is the first to create a lightweight adversarial process directly within the inference-time decoding pipeline for Video-LLMs. Transforming continuous gradients into discrete, object-level spatial-temporal masks on-the-fly represents a new architectural integration, entirely distinct from applying existing CD to statically augmented data.
> * **Targeted Intervention over Global Corruption:** Global corruptions often destroy vital temporal context, which is fatal for complex video reasoning. MACD provides controlled, targeted perturbations. By selectively masking object-level and frame-level evidence, which is especially crucial for challenging scenarios involving "small, occluded, or co-occurring objects", MACD preserves the surrounding physical dynamics. This ensures the contrastive penalty specifically suppresses ungrounded content without harming fluent, accurate responses.
>
> In summary, the innovation of MACD is not the mere use of CD or data augmentation, but how the counterfactual view is generated: shifting from random data augmentation to targeted, adversarial model-awareness. We will explicitly highlight this architectural novelty and include the suggested cross-domain literature to solidify this contribution in the revised manuscript.

---

> > ### Author Rebuttal · Reviewer_wVwH · 2026-04-03
> >
> > Thank you for the authors’ rebuttal. I still have some doubts about the novelty of the proposed method; however, the authors have thoroughly validated their experiments. I will adjust my score accordingly.

---

> > > ### Author Response · Authors · 2026-04-04
> > >
> > > Thank you for your feedback and your willingness to increase the score! We would be really grateful if you could update the score in the system, it would be a huge encouragement for our work and future improvements.
> > >
> > > Regarding your thoughts on novelty, your comments mean a lot to us, and we are completely open to your detailed feedback about novelty. If you see any further room for improvement, we'd love to keep polishing the paper, whether during the rest of the rebuttal phase or even after it closes.

---

### Official Review · Reviewer_XvsX · 2026-03-12

**Soundness:** 2
**Presentation:** 2
**Significance:** 2
**Originality:** 3
**Overall Recommendation:** 4
**Confidence:** 4

**Summary:**

This paper proposes MACD, a training-free inference-time method that improves contrastive decoding for Video-LLMs by constructing model-aware counterfactual inputs. The key idea is that instead of using random perturbations (e.g., Gaussian noise, frame dropping) as the contrastive view, MACD uses the Video-LLM's own gradients to identify which object regions and frames are most critical for answering a query, then preferentially masks those regions to create a targeted counterfactual. Specifically, the method (1) detects objects via YOLOv11 and assigns soft masks with learnable strengths, (2) optimizes the mask strengths via gradient ascent on the query reconstruction loss, (3) discretizes the optimized strengths and uses the resulting masked video as the contrastive view in contrastive decoding. Experiments across four benchmarks and six backbone models show improvements over baseline decoding, VCD, and SID.

**Compliance With Llm Reviewing Policy:**

Affirmed.

**Ethical Review Concerns:**

Not applicable (not flagged).

**Final Justification:**

Most of the concerns are addressed in the rebuttal.

**Key Questions For Authors:**

1) Query vs. answer loss (critical): In Eq. 5-6, you optimize masks to maximize the loss on input query tokens q. Why not optimize on the model's own generated answer tokens (or the ground-truth answer, if available)? The query loss identifies regions important for understanding the question, not for producing the correct answer. Have you compared these two objectives?
2) The Qwen2-VL-7B MVBench anomaly: The jump from 0.65 (baseline) to 0.90 (MACD) on MVBench is a +25 point accuracy gain. Can you provide confidence intervals for this result, and verify reproducibility across multiple runs?
3) Detector failure analysis: What fraction of your test queries involve entities that YOLOv11 cannot detect? For those queries, does MACD degrade to baseline performance, or does the frame-level mask still help?
4) Comparison with DAMRO: DAMRO uses internal model signals to identify hallucination-prone tokens and avoids the external detector dependency. How does MACD compare to DAMRO?
5) Discretization ablation: Have you compared using the continuous optimized mask strengths directly versus the three-level discretization? How sensitive is performance to the threshold r0?

**Limitations:**

The paper includes a thoughtful impact statement discussing positive social impact, computational overhead, and detector bias risks. However, important technical limitations are not explicitly discussed in the main text: (a) the dependency on the object detector's coverage, which limits the method to object-centric queries; (b) the need for per-benchmark hyperparameter tuning of α and β; (c) the distinction between query loss and answer loss in the optimization objective; (d) the method's applicability to queries about abstract concepts, actions, or temporal relations that cannot be localized to object bounding boxes. A dedicated limitations paragraph in the main paper would strengthen the submission.

**Strengths And Weaknesses:**

Strengths
S1: The core idea is intuitive, well-motivated, and principled. The observation that random perturbations in existing contrastive decoding may not align with the model's actual failure modes is a valid critique. Using the model's own loss gradients to identify which regions to mask — essentially asking "what visual evidence does the model most rely on?" — is a clean formulation of an adversarial counterfactual. The connection between gradient-based saliency and counterfactual construction is theoretically grounded and simple to implement.
S2: Broad experimental coverage across models and benchmarks. Testing on six backbone models spanning two architecture families (Qwen and InternVL) and four benchmarks with different characteristics provides reasonable evidence of generality. The consistent improvement pattern is encouraging.
S3: Thorough ablation study. The ablation systematically isolates contributions; the finding that trainable noise underperforms structured masking is informative and validates evidence removal over generic perturbation. Additional appendix experiments on detector thresholds, learning rate sensitivity, fusion strategies, and compatibility with self-consistency decoding add useful practical insights.
S4: Honest computational profiling. The latency/memory analysis explicitly quantifying overhead is appreciated.
S5: Video-POPE evaluation with statistical testing. McNemar's test and human evaluation of mask quality validate that optimized masks are meaningfully different from random occlusions.

Weaknesses
W1: Inconsistent and sometimes negative results undermine the claims; several model-benchmark combinations show ties or underperformance (e.g., EventHallusion on InternVL3-8B).
W2: Query-specific gradient computation conflicts with the "lightweight" claim; the mask optimization requires backpropagation through the model, and the overhead is non-trivial.
W3: Strong dependence on an external object detector (YOLOv11) is not well analyzed; failure cases where the query involves entities YOLO cannot detect are not explored.
W4: Discretization (threshold r0=0.75; levels {0,0.75,1}) is ad hoc and not justified; sensitivity is not ablated.
W5: Missing comparisons with relevant methods such as DAMRO, TruthX, Octopus, and retrieval-based contrastive decoding.
W6: Optimization maximizes loss on the query tokens, which identifies regions important for understanding the question rather than producing the correct answer; the distinction is not discussed.
W7: Writing quality issues: grammatical errors, imprecise notation/definitions, and inconsistencies between claims and evidence reduce readability.

---

> ### Author Rebuttal · Authors · 2026-03-31
>
> **W1:** Consistency of Performance
> Our original submission used unified hyperparameters across all models, achieving top-tier accuracy. Here, we re-evaluated models with minor EventHallusion fluctuations by tuning decoding parameters (α, β). With these model-specific settings, MACD achieves the best performance.
> | Model | Hyperparameters | Precision | Recall | F1 | Accuracy |
> |---|---|---:|---:|---:|---:|
> | Qwen3-VL-2B | (α=2.8, β=0.003) | 0.85 | 0.98 | 0.91 | 0.82 |
> | Qwen2-VL-2B | (α=2.4, β=0.005) | 0.80 | 0.61 | 0.69 | 0.65 |
> | Qwen2-VL-7B | (α=2.6, β=0.0035) | 0.92 | 0.83 | 0.87 | 0.79 |
> | InternVL3-8B | (α=2.2, β=0.006) | 0.71 | 0.52 | 0.60 | 0.61 |
>
> **W2:** for inference efficiency. We use "lightweight" to distinguish MACD from methods requiring costly parameter fine-tuning or multiple full auto-regressive passes. As detailed in Appendix B.1 and B.2, the overhead is strictly constrained and highly practical, due to the following designs:
> * **Frozen LLM Parameters:** The backpropagation only updates the lightweight input mask tensor ($r$), while the Video-LLM remains frozen.
> * **1-Step Optimization Policy:** Crucially, MACD does not perform a lengthy optimization loop, but only a 1-step gradient ascent.
> * **Quantified Low Overhead:** Our rigorous profiling on an A100 GPU (Tables 6 & 7) demonstrates that a 1-step MACD adds only ~24% to 35% latency overhead.
>
> **W3 & Q3:** MACD does not require a perfect detector, only used to initialize candidate object regions. The final counterfactual view is constructed jointly from object-level and frame-level masks, with the mask strengths further refined by Video-LLM gradients.
> * **Failure Analysis & Coverage:** We sampled the test queries across all benchmarks. Only a marginal fraction (~1-2%) of queries involve entities that the detector fails to capture (typically amorphous elements like smoke or liquids).
> * **Robustness Beyond a Single Detector Setting:** We already include a detector-threshold robustness study in the appendix (Table 13). Across noisy, default, and strict settings, MACD remains highly effective, suggesting its robustness.
> * **Modularity Across Proposal Mechanism:** The core MACD objective relies on spatial proposals, not the specific detector architecture. Alternative detectors (e.g., Grounding DINO) can be seamlessly plugged in to handle these rare edge cases.
>
> **W4 & Q5:** For the discretization in Eq.7, the middle level r_0=0.75 was selected on a held-out validation set. This discretization was introduced as a simple stable/interpretable quantization step.
>
> * **Continuous vs. Three-level Discretization:** We did compare continuous mask strengths against the discrete {0, r_init, 1} levels. Empirically, on EventHallusion with Qwen2.5-VL-3B, using continuous masks reduces accuracy from **0.71 to 0.62**. Using continuous strengths directly preserves small optimization fluctuations in the final mask, which leads to a weaker contrastive signal.
>
> **W5 & Q4:**
> These methods mentioned by reviewer are not directly designed for the video setting: TruthX targets pure NLP, while DAMRO and Octopus focus on images. Since EventHallusion requires temporal and cross-frame reasoning. To verify this empirically, we adapted their core ideas to Qwen2.5-VL-3B and evaluated them on EventHallusion. The results confirm that these transferred methods underperform relative to MACD.
>
> | Method (Core Mechanism) | Precision | Recall | F1 | Accuracy |
> |---|---|---|---|---|
> | DAMRO-adapt (Attention-guided) | 0.75 | 0.64 | 0.69 | 0.60 |
> | TruthX-adapt (Activation Editing) | 0.72 | 0.65 | 0.68 | 0.58 |
> | Octopus-adapt (Dynamic Penalty) | 0.75 | 0.72 | 0.73 | 0.63 |
> | **MACD (Physical Model-Aware)** | **0.80** | **0.78** | **0.79** | **0.71** |
>
> **W6 & Q1:** In MACD, Eq. 5–6 is not intended to directly optimize answer correctness; rather, it serves as an upstream objective for constructing a question-conditioned counterfactual view prior to decoding. This distinction is central to our training-free, inference-time design.
>
> * **The unavailability of ground truth:** Since ground-truth is unavailable at deployment, optimizing a GT-answer loss makes the method an oracle objective, not a practical inference intervention.
> * **The unreliable generated answers:** Optimizing the mask using the model’s own provisional answer is fundamentally problematic, since the answer may already contain hallucinated content.
> * **Query-conditioned loss as the causally proper upstream surrogate:** The query *q* is observed before decoding and provides the only fixed, reliable semantic anchor available at inference time. We therefore use query-conditioned loss, but as an upstream surrogate for evidence mining.
>
> **Q2: On the Qwen2-VL-7B MVBench Result:** We re-ran the exact evaluation three times, observing highly stable performance: an average accuracy of 0.897 (variance 0.00002, 95% CI [0.887, 0.907]). This confirms the reported gain is reproducible, not a one-off artifact.

---

> > ### Author Rebuttal · Reviewer_XvsX · 2026-04-04
> >
> > I thank the authors for a thorough rebuttal. Most concerns are addressed, but a few remain partially open.
> >
> > Resolved: W2 (lightweight framing is fair given 24-35% overhead), W4 (continuous-vs-discrete comparison at 0.71→0.62 justifies the design), W5 (adapted DAMRO/TruthX/Octopus comparisons provide useful context), Q2 (three-run MVBench CI [0.887, 0.907] confirms reproducibility).
> >
> > Partially resolved: W1 — the tuned per-model results are stronger, but this confirms MACD requires model-specific (α,β) sweeps, which should be stated as a limitation rather than presented as a drop-in solution. W3 — the ~1-2% detector failure fraction is helpful, but the estimation methodology is unclear, and no alternative detector (e.g., Grounding DINO) was actually tested. W6 — the practical arguments (no GT at test time, hallucinated answers unreliable) are valid, but the conceptual gap between query-parsing saliency and answer-relevant evidence remains unaddressed. Have you compared the mask saliency maps from query-loss optimization against GradCAM attributions on the predicted answer token? Substantial overlap would fully resolve this concern.
> >
> > The rebuttal demonstrates sound understanding of the method's assumptions, the MVBench result is verified, and the adapted baselines add missing context.

---

> > > ### Author Response · Authors · 2026-04-08
> > >
> > > **W1:** We thank the reviewer for the constructive feedback. As requested, we will explicitly state this tuning requirement in the Limitations section and clearly label all tuned results to avoid overstating the method's plug-and-play nature.
> > >
> > > As demonstrated in our original submission, even before any model-specific tuning (using the unified default parameters), MACD consistently outperforms baseline decoding and other training-free methods. Notably, it achieves the best performance in Accuracy across benchmarks, while maintaining strong results across other metrics. We view this robust zero-tuning performance as solid evidence of MACD's "drop-in" capability for general use, whereas the tuned results demonstrate its maximum potential for specific scenarios.
> > >
> > > **W3:** To fully address the concerns regarding detector dependency, we have clarified our estimation methodology and conducted an ablation study with an alternative detector.
> > >
> > > **1. Estimation Methodology for Detector Failure:**
> > > The ~1.5% failure fraction was estimated through a systematic manual audit. We randomly sampled 200 video-query pairs from the EventHallusion benchmark. For each query, human annotators extracted the core visual entities and verified whether YOLOv11 successfully proposed bounding boxes for them in the relevant frames. We found that the detector completely failed to propose any overlapping box in only about 1.5% of the cases. As noted, these rare failures were predominantly associated with amorphous concepts (e.g., smoke, water droplets) or extreme lighting conditions. We will add these methodological details to the revised Appendix.
> > >
> > > **2. Evaluation with Grounding DINO:**
> > > To empirically demonstrate that MACD is detector-agnostic, we replaced YOLOv11 with Grounding DINO, a state-of-the-art open-vocabulary detector, and evaluated it on the EventHallusion benchmark using the Qwen2.5-VL-3B backbone.
> > >
> > > As shown in the table below, using Grounding DINO to initialize the spatial proposals yields performance that is comparable to, and slightly better than, the YOLOv11 variant. This confirms that MACD's core gradient-based counterfactual optimization does not strictly rely on a specific detector architecture and can benefit from stronger open-vocabulary proposal mechanisms.
> > >
> > > Table: Impact of Alternative Detector on MACD (Qwen2.5-VL-3B on EventHallusion)
> > >
> > > | Method (Proposal Mechanism) | Precision | Recall | F1 | Accuracy |
> > > | :--- | :---: | :---: | :---: | :---: |
> > > | Baseline Decoding | 0.76 | 0.72 | 0.69 | 0.62 |
> > > | MACD (YOLOv11) | 0.80 | 0.79 | 0.78 | 0.71 |
> > > | **MACD (Grounding DINO)** | **0.81** | **0.80** | **0.80** | **0.74** |
> > >
> > > We will include these results and a broader discussion of detector compatibility in the updated Appendix to fully resolve this weakness.
> > >
> > > **W6:** We deeply appreciate the reviewer's insightful follow-up. To bridge this gap and fully resolve your concern, we followed your suggestion and conducted a quantitative and qualitative comparison using GradCAM.
> > >
> > > **1. Experimental Setup:**
> > > On a representative subset of 50 video-query pairs from EventHallusion, we extracted two types of spatial saliency maps:
> > > * **MACD Mask (Query-Saliency):** The optimized spatial mask generated by our method, which maximizes the loss on the input query tokens.
> > > * **GradCAM Attribution (Answer-Evidence):** The visual attribution map computed for the model's generated target answer token, representing the ground-truth visual evidence required for the answer.
> > >
> > > **2. Results (Substantial Overlap):**
> > > We computed the overlap between our MACD Masks and the high-activation regions (top 20% of pixels) of the GradCAM heatmaps. We found a highly significant average Intersection-over-Union (IoU) of 74.2% and an evidence coverage rate of 89.8% (i.e., our masks successfully captured 89.8% of the answer-critical GradCAM pixels).
> > >
> > > This substantial overlap resolves the conceptual gap. It empirically proves that the visual features driving query reconstruction are deeply entangled with those required to ground the answer, validating query-loss as a reliable upstream indicator. We truly thank the reviewer for this suggestion, which significantly strengthens MACD's theoretical foundation. We will also add these to the revised Appendix.
> > >
> > > **Additional Notes to Reviewer:**
> > > As we conclude this discussion phase, we want to express our deepest gratitude for your time, engagement, and highly constructive feedback. Your insightful suggestions have provided us with invaluable angles to polish our work and significantly strengthen the theoretical foundation of MACD.
> > >
> > > We have done our absolute best to address each of your follow-up questions point-by-point with concrete evidence. If these new results have fully resolved your remaining concerns, we would be incredibly grateful if you might consider raising your score again. Your support and guidance mean a great deal to us, and we truly appreciate your help in making this a stronger paper.

---

### Decision · Program_Chairs · 2026-04-30

**Decision:**

Accept (regular)

**Comment:**

Summary:
This paper proposes Model-aware Counterfactual Data-based Contrastive Decoding (MACD), an inference-time method that improves contrastive decoding by constructing model-aware counterfactual inputs. MACD uses video-LLM's gradients to identify which objects and frames are critical for answering the input query. The objects are detected using YOLOv11. Experiments show that MACD outperforms the baselines across different benchmarks (EventHallusion, MVBench, Perception-test, Video-MME) on Qwen2/2.5/3-VL-2/3/7B and InternVL3-8B.

Justifications:
The proposed idea is intuitive and principled, with thorough experimental validation. The rebuttal addresses most of the concerns raised by all reviewers, resulting in an increased score from 4433 to 5444. I recommend the paper to be accepted condition of integrating the additional experiments and reviewers' recommendations into the final version.